# Steam Activation of Acid-Chars for Enhanced Textural Properties and Pharmaceuticals Removal

**DOI:** 10.3390/nano12193480

**Published:** 2022-10-05

**Authors:** Tetiana S. Hubetska, Ana S. Mestre, Natalia G. Kobylinska, Ana P. Carvalho

**Affiliations:** 1Faculty of Chemistry, University of Oviedo-CINN, Avda. Julián Clavería, 8, 33006 Oviedo, Spain; 2A.V. Dumansky Institute of Colloid and Water Chemistry, National Academy of Science of Ukraine, Blvd. Akad. Vernads’koho, 42, 03142 Kyiv, Ukraine; 3Centro de Química Estrutural, Institute of Molecular Sciences, Departamento de Química e Bioquímica, Faculdade de Ciências, Universidade de Lisboa, 1740-016 Lisboa, Portugal

**Keywords:** sisal, glucose, acid-mediated carbonization, superactivated carbon, ibuprofen, iopamidol

## Abstract

The present work aims to explore steam activation of sisal or glucose-derived acid-chars as an alternative to KOH activation to prepare superactivated carbons, and to assess the adsorption performance of acid-chars and derived activated carbons for pharmaceuticals removal. Acid-chars were prepared from two biomass precursors (sisal and glucose) using various H_2_SO_4_ concentrations (13.5 M, 12 M, and 9 M) and further steam-activated at increasing burn-off degrees. Selected materials were tested for the removal of ibuprofen and iopamidol from aqueous solution (kinetic and equilibrium assays) in single-solute conditions. Activated carbons prepared from acid-char carbonized with 13.5 M and 12 M H_2_SO_4_ are mainly microporous solids composed of compact rough particles, yielding a maximum surface area and a total pore volume of 1987 m^2^ g^−1^ and 0.96 cm^3^ g^−1^, respectively. Solid state NMR reveals that steam activation increased the aromaticity degree and amount of C=O functionalities. Steam activation improved the acid-chars adsorption capacity for ibuprofen from 20-65 mg g^−1^ to higher than 280 mg g^−1^, leading to fast adsorption kinetics (15–20 min). The maximum adsorption capacities of selected activated samples for ibuprofen and iopamidol were 323 and 1111 mg g^−1^, respectively.

## 1. Introduction

Water scarcity is becoming a reality in an increasing number of countries all over the world, so finding the best solutions to permit the reuse of wastewater is a very important issue for which the contribution of the scientific community plays a central role. Additionally, the contamination of water bodies by emerging pollutants (e.g., pesticides or pharmaceuticals) is also an environmental problem known for quite a long time [1,2] resulting from the lack of infrastructures or the inefficacy of the conventional wastewater treatment processes. This issue boosted the search for processes that could be introduced in the water treatment facilities for their upgrade to allow the removal of recalcitrant compounds. Technologies based on adsorption onto activated carbons are an example of those processes which also present the advantage of being easily implemented in the existing plants [3], as proved in a full-scale addition of powdered activated carbon to the activated sludge reactor of an urban wastewater treatment plant, during the recently developed LIFE Impetus project [4]. The data gathered not only through academic studies [5,6,7,8,9] but also when these processes were implemented in the wastewater treatment plants prove that this is a viable method to improve effluents quality, namely regarding the removal of pharmaceutical compounds [4,10,11]. Therefore, our group has been interested in testing new precursors and alternative preparation methodologies to obtain highly porous activated carbons that could be considered candidates for the removal of recalcitrant compounds. This is a very important topic, especially if the methodology involves the valorization of renewable biomass residues (e.g., pine nut shell and pine cone [8], carob processing residues [9], cork processing by-products [12] or yeast biomass [7]). This research area addresses the circular economy principles: (i) by decreasing waste generation while producing valuable products; (ii) ideally in a facility close to the area where residues are generated; (iii) by allowing to face demanding needs for high-performing nanoporous materials in environmental remediation technologies and energy storage and production.

Frequently the synthesis of activated carbon materials comprises two heating steps: Carbonization to increase the carbon content of the precursor followed by activation to develop the nanoporous network [13,14]. Pyrolysis is the conventional carbonization process used at industrial scale, and in research studies, consisting in carbonizing the biomass residues under an inert atmosphere at around 400 °C or higher temperature. An alternative carbonization route, that has been explored mainly since 2001, is hydrothermal carbonization (HTC) [15]. This synthesis process is inspired by the natural coal formation, as it is made under self-generated pressure since the synthesis occurs in an autoclave, using water as solvent, and temperatures between 100 °C and 350 °C, being the obtained carbon materials designated hydrochars [5,14]. More recently proposed, and far less explored, acid-mediated carbonization (AMC) is also a viable procedure to transform carbon-containing precursors into a solid carbon-rich material still containing relevant percentages of oxygen [14,16]. AMC consists of the acid-catalyzed polycondensation of carbon-containing precursors at atmospheric pressure and temperatures, in general, around 100 °C under air, even though some studies report temperatures as high as 800 °C under nitrogen flow. The process yields a solid material that can be denominated as acid-char. When using a solid biomass precursor, the AMC is preceded by an acid digestion step which will allow the breaking of the bonds of the two main biomass components, the ether bonds between hemicellulose monomers, and the glucosidic bonds between glucose units composing cellulose [17]. The resulting acidic black liquor is rich in saccharic units which via the dehydration, polymerization, and carbonization catalyzed by sulfuric acid will yield an acid-char [18].

AMC is an interesting carbonization route to maximize the availability of cellulose and hemicellulose present in carbon precursors with high inorganic content, as demonstrated by the studies of Wang et al. [18,19]. These authors explored the AMC of rice husk (18.2 wt.% hemicellulose, 35.2 wt.% cellulose, 24.5 wt.% lignin, and 18.8 wt.% silicon dioxide) with H_2_SO_4_ at atmospheric pressure and temperature lower than 100 °C. The results proved that it was possible to activate chemically the rice husk-derived acid-chars and to obtain activated carbons with micropore networks and BET area of 2500 m^2^ g^−1^ in the case of KOH, and micro-mesopore materials with BET areas between 750 and 2700 m^2^ g^−1^ when H_3_PO_4_ activation was used.

The AMC is also an efficient route for carbonizing liquid carbon precursors, as shown by Cui and Atkinson [20], which successfully prepared glycerol-derived acid-chars at temperatures in the range of 400 °C to 800 °C. Depending on the acid catalyst (H_2_SO_4_ and/or H_3_PO_4_), the obtained oxygen-rich chars contain sulfur (0.43–4.20 wt.%) or phosphorus (6.32–16.31 wt.%) groups, that enhance the reactivity for subsequent physical activation with steam or CO_2_ allowing to gather a well-developed hierarchical pore network (1000–2400 m^2^ g^−1^) with heteroatom doping.

In a previous study, we reported the synthesis of sisal-derived acid-chars showing that, by controlling the H_2_SO_4_ concentration during the digestion and polycondensation steps, it is possible to transform low-density biomass into versatile and tailored acid-chars which were further chemically activated with KOH and K_2_CO_3_ [21]. The control of the AMC parameters allowed to obtain sisal-derived acid-chars with distinct morphologies and densities, and depending on the selected acid-char, chemical activating agent, contact method and, activation temperature, further activation yielded exclusively microporous or micro-mesoporous (50:50) materials with BET area values up to 2300 m^2^ g^−1^.

In the present study, we explore the steam activation of acid-chars prepared from sisal (Agave sisalana) and glucose aiming to evaluate the effect of the precursor and of the H_2_SO_4_ concentration during AMC on the properties of the nanoporous materials obtained by steam activation at increasing burn-off degrees. Steam activation of the selected acid-chars allowed to obtain superactivated carbons with BET area values close to 2000 m^2^ g^−1^ and nanoporous networks that resemble those usually attained with KOH activation of hydrochars [22,23,24,25,26,27] or acid-chars [21], in any case at high KOH:hydrochar weight ratios (3:1 to 5:1). Selected acid-chars and derived superactivated carbons were successfully tested for the adsorption of two pharmaceutical compounds—ibuprofen and iopamidol—from aqueous phase and were benchmarked against commercial counterparts and high performing activated carbons obtained by KOH activation of acid-chars or hydrochars.

## 2. Materials and Methods

### 2.1. Synthesis of Acid-Chars 

The precursor used was sisal (*Agave sisalana*) residues supplied by a rope manufacturing company (Cordex, Portugal) and glucose (98 %, Aldrich, St. Louis, MO, USA). The polycondensations of both precursors were performed using different concentrations of H_2_SO_4_ (13.5 M, 12 M, 9 M) at 90 °C for 6 h and, in the case of sisal (S), it was preceded by the digestion step to extract the saccharic units present in the sisal fibers. For this, 4 g sisal were digested under stirring in 40 mL H_2_SO_4_ solutions (95–98%, Sigma-Aldrich, St. Louis, MO, USA) at 50 °C for 15 min in the case of 12 M and 13.5 M solutions, and for 1 h in the case of the 9 M solution. For glucose (G), the process started by dissolving 4 g in 40 mL of the desired H_2_SO_4_ solution for 10 min under stirring. The synthesis temperature was controlled by a water bath (Model 1201, VWR Scientific, Radnor, PA, USA). After the polycondensation/carbonization was processed, the solid products—acid-chars—were collected by filtration, washed with distilled water until neutral pH, dried at 100 °C overnight, and finally milled to obtain particles with dimensions smaller than 74 μm. 

The sisal and glucose-derived acid-chars were labeled as SX and GX, respectively, where X corresponds to the H_2_SO_4_ concentration (M) used (e.g., S13.5).

The acid-char preparation yield was determined by Equation (1):(1)$$Acid-char\;preparation\;yield,\;\%=\frac{m_{\mathrm{acid}-\mathrm{char}},\;\;\;g}{m_{\mathrm{precursor}},\;\;\;g}\times 100\%$$

### 2.2. Activation of Acid-Char Samples

The physical activation with steam followed the experimental setup developed in previous works [8,9,12]. The acid-char samples were introduced in a quartz reactor placed in a vertical furnace (Thermolyne, model 21100, Waltham, MA, USA). The steam was generated in a bubbler half full of distilled water heated at 70 °C and carried to the sample by N_2_ flow (5 cm^3^ s^−1^). The acid-char (≈1 g) was heated (10 °C min^−1^) until the desired activation temperature (850–925 °C) and kept for 1 h, after which the steam flow was turned off and the sample was cooled to ambient temperature under inert atmosphere. The activated carbons obtained were designed by adding the activation “st” and the burn-off value (%) to the name of the acid-char precursor (e.g., S13.5st88 is the sample obtained with 88 % burn-off after the steam activation of the acid-char prepared from sisal using a 13.5 M H_2_SO_4_ solution).

The “burn-off” degree (Burn-off, %) of the obtained activated carbon samples was determined using Equation (2):(2)$$Burn-off,\%=100-\left(\frac{m_{\mathrm{activated}\;\mathrm{carbon}},\;g}{m_{\mathrm{acid}-\mathrm{char}},\;g}\right)\times 100\%$$

For direct comparison with acid-char preparation yield, the activation yield was also calculated by Equation (3):(3)$$Activation\;yield,\;\%=\frac{m_{\mathrm{activated}\;\mathrm{carbon}},\;g}{m_{\mathrm{acid}-\mathrm{char}},\;g}\times 100\%$$

### 2.3. Characterization of the Materials

The morphological and textural characteristics of the carbon materials were obtained from adsorption-desorption isotherms of N_2_ and CO_2_, powder X-ray diffraction (XRD), Fourier transform infrared (FTIR) spectroscopy, scanning electron microscopy (SEM), and thermal analysis. The N_2_ adsorption/desorption isotherms at −196 °C were obtained in a Micromeritics ASAP 2010 apparatus (Norcross, GA, USA), while the CO_2_ experiments at 0 °C were made in a Micromeritics ASAP 2020 (Norcross, GA, USA). Before adsorption experiments, the samples (50–60 mg) were outgassed overnight at 120 °C under a vacuum better than 10^−2^ Pa. Powder XRD data were obtained at room temperature using a PAN Analytical PW3050/60X’Pert PRO (Philips, Almelo, The Netherlands) equipped with an X’Celerator detector and with automatic data acquisition (X’Pert Data Collector (v2.0b) software). The diffractograms were collected at room temperature using Ni-filtered CuKα (λ = 1.5406 Å) monochromatized radiation as incident beam (40 kV–30 mA). The size and morphology of the samples were analyzed on the scanning electron microscope JSM-6100 (JEOL, Tokyo, Japan) operating at 20 kV.

FTIR spectra were recorded from the samples pressed in pellets with KBr using a Bruker Tesor 27 spectrometer (Billerica, MA, USA). Thermal gravimetric analysis (TGA) measurements were performed in a Mettler-Toledo TGA/SDTA851 SDTA851 (Columbus, OH, USA) thermobalance (N_2_ atmosphere, 10 °C min^−1^). The chemical composition of samples was obtained by elemental analysis (C, H, N, S) using LECO CHNS-932 (St. Joseph, MI, USA).

Single pulse ^13^C MAS NMR spectra were recorded with an Avance III 400WB Bruker spectrometer operating at a carbon frequency of 100.61 MHz and a spinning frequency of 5 kHz. The 90° pulse width was 4.5 ms and high-power proton decoupling was performed during the recording of the spectra. The recycle delay was 20 s and 4000 scans were added for every spectrum.

The pH at the point of zero charge (pH_PZC_) was determined by reverse mass titration following the procedure proposed by Noh and Schwarz [28]. Briefly, a 10% suspension of the material previously dried was mixed with decarbonized deionized water and kept under stirring for, at least, 24 h under N_2_ atmosphere at room temperature. After this period, the pH was measured with a Symphony SP70P pH meter (VWR, Radnor, PA, USA). The pH values for lower solid weight fractions were obtained by successive dilution of the initial suspension (8, 6, 4, 2, and 1 wt.%, respectively). The pH_PZC_ value corresponds to the plateau of the curve of equilibrium pH versus solid weight fraction.

Apparent tapped densities of the acid-chars and activated carbons were determined through a methodology adapted from the literature for powdered materials with 90% of particles with dimensions lower than 0.177 mm (80 mesh) [29,30]. Typically, 0.5 g of powdered activated carbon (model AB204-S/FACT, Mettler Toledo, Columbus, OH, USA)) was filled in a graduated cylinder and tapped in a rubber pad until constant volume. The apparent densities were determined by Equation (4):(4)$$density,\;kg\;m^{-3}=1000\times \frac{\mathrm{weight}\;\mathrm{of}\;\mathrm{sample},\;g}{\mathrm{volume}\;\mathrm{of}\;\mathrm{sample},{\;\mathrm{cm}}^{3}}\times \frac{\left(100-moisture,\;\%\right)}{100}$$

### 2.4. Batch Adsorption Experiments

Batch adsorption experiments were carried out in 50 mL glass flasks containing 30 mL of stock solutions and around 6 mg of adsorbents. The stock solutions of ibuprofen (sodium salt, Lot BCBC9914V, Sigma-Aldrich, St. Louis, MO, USA) or iopamidol (Lot 163926HQ01324 Hovione, Loures, Portugal), presenting pH = 5.5, were prepared with deionized water obtained from Milli-Q purification systems without pH adjustment. The samples were stirred at 700 rpm (multipoint agitation plate Variomag Poly, Port Orange, FL, USA) at 30 °C controlled with awater bath equipped with a temperature controller (model GD100, Grant Shepreth, UK). After a predetermined contact time, the suspensions were filtered using Nylon membrane filters of 0.45 µm pore size. The concentration of the recovered solution was determined by UV–vis spectrophotometry (Genesys 10S spectrophotometer, ThermoFisher, Waltham, MA, USA) at a wavelength corresponding to the maximum absorbances at 222 nm (ibuprofen), and 242 nm (iopamidol).

Adsorption kinetic and isotherm experiments were conducted to examine the performance of the synthesized carbon materials towards the removal of the two pharmaceutical compounds. To measure the adsorption kinetics, 180 mg L^−1^ analytes solutions were used, and adsorption was measured at time intervals ranging from 3 min to 24 h. Adsorption isotherms were obtained using 0.2 g L^−1^ of activated carbon with different concentrations of ibuprofen and iopamidol aqueous solutions between 30 mg L^−1^ and 400 mg L^−1^ at natural pH assuring 24 h of contact time. The concentration of analytes remaining in solution at equilibrium was determined as above-mentioned.

## 3. Results and Discussion

### 3.1. Characterization of Acid-Char and Activated Carbon Samples

The yield of activated carbons’ production is an important parameter when optimizing the preparation conditions of new materials. As shown in Figure 1a, the yields of acid-chars preparation are markedly dependent on the concentration of H_2_SO_4_ in line with the data reported in a previous study [21]. Regardless of the carbon precursor (sisal or glucose), the values range between 32–38% for 12 M or 13.5 M H_2_SO_4_, and below 15% for 9 M H_2_SO_4_. 

Regarding the activation step (Figure 1b), as expected, higher activation yields were obtained when lower temperatures were used or, in other words, when the process resulted in lower burn-off degrees. Furthermore, the data show that activation yield does not depend on either the carbon precursor or the H_2_SO_4_ concentration used to prepare the acid-char. In fact, all the materials activated at 850 °C present activation yields between 27% and 31%.

The results depicted in Figure 1a also show that the apparent density of the acid-chars is independent of the carbon precursor, being the results correlated with H_2_SO_4_ concentration, as already reported in a previous study focused only on sisal [21]. The activation leads to the decrease of the apparent density (Appendix A), this being the expected trend considering the porosity development during the activation process, which also explains the higher density generally achieved at lower burn-off degrees (Figure 1b). Comparing the apparent density of activated carbons obtained at burn-off degrees between 80–82% (i.e., S13.5st80, S12st82, G12st81, S9st80, and G9st81), it is possible to verify that the values reflect mainly the effect of the H_2_SO_4_ concentration used to synthesize the acid-char. Only in the case of activated carbons prepared from S13.5 (the acid-char with the highest density), no significant effect of burn-off on the apparent density was observed. Lastly, it must be noted that the steam activation of the acid-chars prepared in this study allowed to prepare activated carbons with apparent densities such as those reported for the K_2_CO_3_ activation of acid-chars S12 or S13.5, and higher than those obtained by KOH activation [21].

The elemental composition of the acid-chars and derived activated carbons are presented in Table 1, along with the values of the pH_PZC_. The results show that all the acid-chars are composed of 55–62% of carbon that rises up to 90–94% after activation. Consequently, the percentage of heteroatoms in the activated carbons is low, with oxygen continuing to be the major heteroatom present in the structure. Contrary to sulfur and oxygen, the nitrogen percentage increases upon activation, pointing out that nitrogen is present in less thermolabile groups than oxygen or sulfur moieties.

The morphological characteristics of the carbon materials were assessed by SEM microscopy (Appendix A for acid-chars, and Figure 2 for activated carbons). Regardless of the precursor, acid-chars prepared with 13.5 M and 12 M H_2_SO_4_ are composed of irregular, dense and sharp particles while those prepared with 9 M H_2_SO_4_ present an aerogel-like structure. The steam activation allows to maintain the morphology of the pristine acid-char (Figure 2), which is in line with the slight decrease of the apparent density after steam activation observed in all the cases (Appendix A). 

XRD patterns of the acid-chars (Appendix A) show that the concentration of H_2_SO_4_ in the digestion step, as well as the nature of the precursor, have a low impact on the structural properties of the materials. The acid-chars have an amorphous-like carbon structure with a low crystallinity, as evidenced by the broad and low-intensity diffraction peaks at 20–23 °2 θ and 41–45 °2 θ, in line with a previous study [21]. Regarding activated carbons, no major changes were observed in the XRD patterns (data not shown), even though the steam activation was made at temperatures up to 925 °C.

The N_2_ adsorption–desorption isotherms at −196 °C (Appendix A) show that all acid-chars present type II isotherms according to IUPAC classification [31], pointing out that these are nonporous materials with external surface area resulting from the aggregation of finely divided particles. In fact, the acid-chars prepared with 9 M H_2_SO_4_, which have an aerogel-like structure composed of aggregated particles, are those presenting the highest and sharpest N_2_ uptake at relative pressures close to the unit. Accordingly, these materials present a low surface area (10–15 m^2^ g^−1^). 

Figure 3 presents the N_2_ adsorption isotherms of some representative examples of the activated carbons prepared from sisal- and glucose-derived acid-chars. All the curves are type I isotherms [31], with an initial sharp N_2_ adsorption followed by a plateau characteristic of essentially microporous materials. The effect of the burn-off on the texture development is exemplified by the isotherms of the carbons derived from acid-char S13.5 (Figure 3a) that, as expected, show that increasing burn-off degree led to the progressive increment in the N_2_ adsorption capacity. The isotherms obtained with activated carbons prepared from S9 or G9 acid-chars (aerogel-like morphology) along with the characteristics of type I curves present an increase of N_2_ uptake at relative pressure close to 1. This feature is attributed to the aggregation of fine particles (external surface area), in line with previous observations in the N_2_ adsorption isotherm of the G9 acid-char.

Table 2 summarizes the textural properties of the samples. The specific surface area (*A*_BET_) was determined through the BET equation according to the IUPAC recommendation for microporous materials [31,32]. The microporosity of activated carbons was analyzed applying the α_s_-method using the reference isotherm reported by Rodríguez-Reinoso et al. [33]. Total pore volume (*V*_total_) was evaluated through the Gurvich rule [34].

The synthesis protocol herein explored allowed to obtain nanoporous carbons with *A*_BET_ between 1100 m^2^ g^−1^ and almost 2000 m^2^ g^−1^. Appendix A shows that there is a linear relation (*R*^2^ > 0.92) between the *A*_BET_ values and the burn-off of the activation step, regardless of the carbon precursor or the acid-char synthesis conditions. On the other hand, and in line with the analysis of the N_2_ isotherms configuration, textural parameters confirm the predominant micropore nature of all the materials. Whenever the acid-char was prepared with 12 M or 13.5 M H_2_SO_4_ the mesopore volume (*V*_meso_) of the resulting activated carbon is, at the maximum, 10% of the total pore volume, while for materials prepared from S9 or G9 acid-chars the percentage of *V*_meso_ reaches 20% or 30%, respectively. Regarding microporosity, while for samples with burn-off of around 70% there is a 50:50 distribution of ultra and supermicropore volumes; as the burn-off increases, the volume of supermicropores became predominant, being 100% or close, for burn-off higher than 80%. This analysis agrees with the pore size distributions presented in Figure 3b,d where the gradual increase of large micropores is clearly observed, along with some small mesopores with widths up to 3.0 nm in the case of material S13.5st87.

Except for S13.5-derived carbons, the expected inverse relationship between the apparent density and the micropore volume (see Appendix A) is observed. In fact, only the steam activation of S13.5 at increased burn-offs allowed to continuously increase the porosity development, keeping the apparent density almost constant. This result points out the higher structural stability of the acid-char S13.5 in comparison with that of sample S12 which presents only a slightly lower apparent density. 

It is important to notice that the textural properties of the steam-activated materials herein prepared compare favorably with those obtained by chemical activation of S13.5- and S9-derived acid-chars reported in a previous study, in particular, those KOH activated at a 3:1 weight ratio [21]. This achievement is very important since it is known that chemical activation with high amounts of KOH is a very hazardous process with a high economic cost [35,36]. In the literature, most superactivated carbons with textural features such as those herein reported were prepared from KOH activation of hydrochars [22,23,24,25,26,27], aiming for their use as supercapacitors or for CO_2_ capture. Lastly, it must be stressed that the present synthesis methodology is an alternative route that is especially interesting in the case of biomass residues with high inorganic content since the initial H_2_SO_4_ digestion step enables to recover only the carbon content of the biomass which is further carbonized under acid medium [14,16,19]. 

The microporosity of selected acid-chars and activated carbons tested in the liquid phase assays was further characterized through CO_2_ adsorption–desorption experiments (Figure 4a,c). The micropore size distributions (MPSD) obtained by the method proposed by Pinto et al. [37] are presented in Figure 4b,d. This method is based on the Dubinin–Radushkevich (DR) equation and uses the integral adsorption equation thus, contrary to the Dubinin–Radushkevich–Stoekli (DRS) equation, does not limit the shape of the distribution to a presumed shape. Acid-chars present almost monomodal MPSD mainly composed of ultramicropores (Figure 4b). The distributions show that while acid-char S13.5 has the narrowest MPSD, samples S9 and G9 present progressively wider distributions with a small portion of micropores with widths larger than, respectively, 1.5 nm and 1.7 nm. As expected, upon activation MPSDs are shifted towards larger micropores (Figure 4d). All the activated materials analyzed present a maximum centered at 0.7 nm, and while materials S9st80 and G9st81 have a continuous micropore size distribution up to 2.0 nm, sample S13.5st87 has a marked bi-modal distribution with the highest percentage of supermicropores. These results are in accordance with the microporosity characterization through the N_2_ adsorption data, which revealed the predominance of (or even only) a supermicropore system (see Table 2).

The thermal stability of samples was characterized by TGA (Figure 5). According to thermogravimetric (TG) profiles, the use of various concentrations of H_2_SO_4_ has a low influence on the thermal stability of the samples.

In the case of the acid-chars, and in line with previous data for sisal-derived acid-chars [21], TG curves show two major weight loss steps with corresponding thermal events in the DTG curves (Figure 5a). Up to 150 °C samples lost between 5 and 12% of weight due to the release of physically adsorbed water, pointing out some hydrophilicity of the materials. This is not surprising since pH_PZC_ values of the acid-chars are between 2 and 3 (Table 1). The second mass loss (from 150 °C up to 900 °C) can be assigned to the decomposition of the surface groups formed during the polycondensation process. According to the literature [38], we can assume that the more labile groups decomposed in this step may be carboxylic acids and lactones, which are degraded at temperatures up to 400 °C and 650 °C, respectively. Other surface groups such as phenol, carbonyl, quinones, hydroquinones, and ethers suffer degradation mainly between 600 and 700 °C.

The thermal analysis of selected activated carbons with burn-off between 68% and 87% is shown in Figure 5b. For samples with burn-off ≈ 80%, the total weight loss was only 9%, proving the high thermal stability of the materials. At a temperature below 100 °C (first step), the weight loss is associated with the volatilization of physically adsorbed water. Regarding the decomposition of carbon S13.5st68, this first step covered a wider temperature range (below around 150 °C) with greater weight loss (≈18%), which can be attributed to the more intense interaction of water molecules with the carbon surface groups, which seems to indicate the presence of more abundant oxygen-containing functional groups in this sample. The weight loss at T > 600 °C may be assigned to the decomposition of surface functionalities such as lactones, phenol, carbonyl, quinones, hydroquinones, and ethers groups. Finally, it cannot be disregarded that some weight loss of both acid-chars and derived activated carbons is certainly associated with the decomposition of nitrogen and/or likely sulfur functionalities.

^1^H and ^13^C-NMR spectroscopies with magic angle spinning (MAS) are sensitive to the surface functional groups of various solids [30]. The solid-state NMR technique was employed to explore the direct information concerning surface elements of carbon materials with a high number of functional groups according to TGA data. ^13^C MAS NMR spectra of the starting acid-char (S13.5) and corresponding activated carbon S13.5st68 are shown in Figure 6.

These spectra have several broad intense peaks and several signals which can be attributed to carbon nuclei in different environments. The spectrum of acid-char S13.5 (Figure 6a) consists of several overlapping located at approximately 10; 25; 34; 56; 75; 90; 118; 160; 190 and 290 ppm. Generally, signals in the range 0–90 ppm are assigned to sp^3^ hybridization carbon, as well as between 0–45 ppm, this includes CH_3_; CH_2_; and CH-groups [39]. The peak at 34 ppm can be attributed to nanodiamond carbon [39]. The signal for carbon alkoxy groups appears at 56 ppm. All bands from 50 to 90 ppm are ascribed to sp^3^ carbon connected to oxygen bonded to heteroatoms such as nitrogen or sulfur (e.g., alcohols; amines; ethers; CH*_x_*(NO or NO_2_) or CH*_x_*–SO_2_–R), confirming that besides nitrogen from the precursor also sulfur was incorporated in the architecture of acid-char. In the sp^2^ spectral region, the sum of bands spanning from 100 to 160 ppm is assigned to aromatic carbon. This range includes oxygen-containing aromatic carbon—phenols (Ar–OH) and phenyl ethers (Ar–O–R). Note that the sideband intensity at 290–190 and 20–0 ppm is added to the quaternary aromatic carbon fraction. Finally, the sum of all bands between 160 and 220 ppm is assigned to C=O, i.e., carboxyl and ketone functional groups. Additionally, the evaluation of the integral intensity of the ^13^C MAS NMR peaks of S13.5 acid-char results in a total amount of carbons connected to heteroatoms ca. 1.5%, which agrees with the total content of heteroatoms (N, S) determined by element analysis (1.56%, Table 1).

The ^13^C MAS NMR spectrum of the S13.5st68 activated carbon sample shows a similar appearance with very broad intense peaks (Figure 6b). The spectrum consists of four main broad components (−7.6, 70, 117 and 173 ppm) and a smaller band at 235 ppm. The strongest peak at 117 ppm is caused by aromatic C=C groups [40]. The strong signal at 173 ppm is attributed to C=O, whereas the band at the lowest ppm values can be assigned to aliphatic C–C bonding groups. One can also notice the appearance of a low intense peak at 70 ppm, of which assignment is made following residual C–S groups of the sample. The ^13^C MAS NMR spectrum of the activated carbon sample shows that the acid-char was transformed into a polycyclic material with the preponderance of aromatic structures with C=O-containing groups, as indicated by the signals observed in the 80–140 ppm and 150–200 ppm regions (Figure 6b). Thus, according to NMR data, the aromaticity degree considerably increased with the activation along with the C=O functionalities (carboxyl and ketone groups), even for the less activated carbon of the S13.5 series, i.e., S13.5st68 sample.

FTIR spectra of the acid-chars presented in Figure 7a reveal that regardless of whether sisal or glucose are used as carbon precursors similar results are obtained, resembling the data reported in the previous study focused on sisal-derived materials [21]. In fact, all the spectra show a strong, broad absorption band at 3600–3300 cm^−1^, characteristic of the stretching vibration of hydroxyl (O–H); peaks at 2850, 2927, and 1451 cm^−1^ attributed to CH-stretching; an intense band at 1694 cm^−1^ assigned to C=O axial deformation; a band at 1591 cm^−1^ attributed to aromatic compound vibrations; a band at 1400 cm^−1^ correspondent to symmetrical angular deformation on the plane of the methylene groups; a band around 1137 cm^−1^ assigned to C–O–C asymmetrical axial deformation of ethers or C–(C=O)–C axial; and angular deformation of ketones (which commonly appears at 1300–1100 cm^−1^). Furthermore, a well-defined band was observed at 1119 cm^−1^, which is a characteristic S–O vibration of SO_3_^2−^ moieties in the samples [21,41]. Therefore, bands assigned to oxygen functionalities and aliphatic and aromatic carbon, along with bands assigned to sulfur functionalities were identified in the FTIR spectra of acid-chars.

Figure 7b presents the FTIR spectra of activated carbon revealing that even after the high-temperature treatment the materials retain oxygen functionalities as proved by the band assigned to O–H stretching of surface hydroxyl groups (3640–3200 cm^−1^), the peak assigned to stretching vibrations of C=O (~1640 cm^−1^) in lactones, anhydrides, ester, or carboxylic acid and quinones [42]. The band centered at 1094 cm^−1^ can be attributed to C–OH stretching in aromatic or aliphatic alcohols [42] or also to sulfonic functionalities [21,41]. The peaks at 2919 cm^−1^ and 2847 cm^−1^ are assigned to, respectively, symmetric and asymmetric C-H stretching vibrations of aliphatic moieties in all the activated carbon materials analyzed. The peak at 599 cm^−1^ can be assigned to out-of-plane deformation vibrations of C–H groups at the aromatic plane edges [43]. 

The acid-base features of the materials were also assessed by the determination of pH_PZC_, measured by the reverse mass titration method [28,44]. The pH_PZC_ values of selected activated carbons (Table 1) indicate that activation has, as already shown by TG/DTG and FTIR data, an important impact on the surface chemistry of the samples. The surface of the acid-char has acidic pH (pH_PZC_ ≈ 2), while after steam activation the pH_PZC_ value increased to 5.7–6.2, showing the decrease of acidic groups upon activation.

### 3.2. Ibuprofen and Iopamidol Adsorption onto Carbon Adsorbents

Selected acid-chars and activated carbon samples were evaluated for their capacity to remove pharmaceuticals from an aqueous solution. The probe molecules chosen for this study were ibuprofen (anti-inflammatory) and iopamidol (iodinated contrast media), two pharmaceuticals largely used worldwide and commonly detected in the aquatic environment. The structure and some physico-chemical properties of these two pharmaceuticals, important to rationalize the liquid phase adsorption results, are gathered in Table 3.

The results of the screening assays testing ibuprofen removal (Figure 8) confirm the paramount importance of the nanoporous network for extensive adsorption of this anionic compound. While with 12 mg of acid-chars uptakes between 20 and 65 mg g^−1^ were achieved, under the same experimental conditions only with 6 mg of activated carbons values higher than 280 mg g^−1^ were attained. To rationalize these results, we must consider both the effect of the surface chemistry and the textural properties of the materials along with the molecule features.

Regarding the influence of surface chemistry, no attractive electrostatic interactions are expected between the ibuprofen anion and the negatively charged surface of the acid-chars (solution pH > pH_PZC_). However, for activated carbons there may be some contribution of positive electrostatic interactions since, in this case, the surface is positively charged or neutral (solution pH < pH_PZC_). So, the results are mainly linked with the nanopore system that is much more developed in the case of activated carbons, as it was proved by the pore size distributions obtained from the CO_2_ adsorption isotherms and textural parameters obtained from N_2_ adsorption data. In fact, while no diffusion constraints are expected for the activated carbons, since most of the micropore network (Figure 4d and Table 2) is above the ibuprofen critical dimension (0.72 nm, Table 3), for acid-chars only a small fraction of the micropore network (Figure 4b and Table 2) is available for the ibuprofen anion. Therefore, in the case of activated carbons both surface chemistry and texture favor the ibuprofen anion uptake.

Considering the results of the screening assays activated carbons S13.5st87, S9st80, and G9st81 were chosen for kinetic and equilibrium studies with ibuprofen and iopamidol.

The kinetic curves of ibuprofen adsorption, reproduced in Figure 9a, are almost coincident showing a very marked decay in the first minute. After 5 min, the adsorption proceeds more slowly towards equilibrium which was attained after 20 min. For iopamidol adsorption, the kinetic profiles (Figure 9b) reveal that after the first minute slower adsorption continues until reaching equilibrium only after 6 h, with the uptake following the trend S13.5st87 > G9st81 > S9st80.

The experimental data were fitted to the pseudo-first and pseudo-second order kinetic models [45,46] with better determination coefficients (higher than 0.9967) for the pseudo-second order model. The kinetic parameters, listed in Table 4, corroborate the previous analysis of the kinetic profiles revealing that the adsorption processes of both compounds are very quick phenomena, with the major difference being the uptake of the various systems. In the case of ibuprofen, the similar removal points out that under the experimental conditions used the difference in the textural parameters is not reflected in the behavior of the material. On the contrary, the increase of supermicropore and mesopore volumes has a positive effect on iopamidol uptake, certainly because at the solution concentration used in iopamidol assays, besides single molecules (critical dimension 0.6 nm), this compound is also present as aggregates, namely, dimers and trimers with critical dimensions of 1.2 and 1.8 nm, respectively [47].

Since we are comparing the uptake of species with quite different molecular weights, it is relevant to confront the adsorption quantities express in chemical quantity (mmol g^−1^). The values listed in Table 4 reveal that, regardless of the material, the number of ibuprofen moles adsorbed is around 50% higher than that of iopamidol, as was expected considering the molecular dimensions of these two pharmaceuticals that favor its packing in the pore system.

Based on the kinetic data, samples S13.5st87 and G9st81 were selected to proceed with the equilibrium study. The isotherms obtained are presented in Figure 10 revealing that the trend of the uptakes achieved on the kinetic assays is valid for a larger equilibrium concentration range. In fact, the ibuprofen adsorption isotherm profiles are coincident, and, once again, for iopamidol S13.5st87 presents the highest adsorption capacity. Experimental data were fitted to the Langmuir [48] and Freundlich [49] models revealing better adjustment to the former model (lower *χ*^2^ values, Table 5) in accordance with the fitting curves displayed in Figure 10.

Confronting the present data with that reported in a previous study based on chemical activation of sisal-derived acid-chars [21], it can be concluded that, besides identical textural characteristics, also similar performance for the removal of ibuprofen and iopamidol was obtained. So, it can be concluded that by using steam activation it was possible to mimic the micropore system of KOH activated acid-chars [21]. Also, the comparison of the adsorption capacity of these samples with those obtained in previous studies using commercial activated carbons for wastewater treatment strengthens the conclusion that the methodology followed allowed the preparation of materials with exceptional performance for the adsorption of the two model compounds tested. In fact, the adsorption capacity achieved is one order of magnitude higher than that presented by the commercial samples for iopamidol [47] and doubles that for ibuprofen [12].

The characteristics of the samples prepared in this study can also be compared with activated carbons prepared from sucrose-derived hydrochars [23]. When KOH activation was used also highly developed microporosity was obtained with similar pore size distribution despite the different mechanisms of the activation processes.

It is important to stress that, to the best of our knowledge, the herein proposed synthesis methodology is an alternative to the protocol generally followed to prepare superactivated carbons, which implies the use of high weight ratio KOH:carbon precursors. For example, sucrose-derived hydrochars activated with KOH attained a surface area of 2400 m^2^ g^−1^ and 95% of pore volume being supermicropores, but at the expense of using a 4:1 KOH:hydrochar weight ratio [23] and rice-husk-derived acid-chars yielded a material with 2488 m^2^g^−1^ but only after being activated with a 5:1 KOH:acid-char weight ratio [18].

Moreover, it is not common to find literature data on steam activated materials gathering, as herein reported, surface area around 2000 m^2^ g^−1^ and total pore volume > 0.8 cm^3^ g^−1^, composed mainly of supermicropores.

Comparing the properties of the materials synthesized in the present study with those reported in the literature highlights the importance of the characteristics of the char, that, in turn, are linked not only to the carbon precursor but also to the carbonization methodology. In fact, acid-mediated carbonization, besides allowing to recover the carbon content of biomass with a high inorganic amount, and to process liquid or wet biomass, yields acid-chars with designed properties (and morphologies), namely controlling the acid concentration [14,16]. The results obtained prove that acid-mediated carbonization with 13.5 M H_2_SO_4_ leads to acid-chars with low ash content, compact structures with high density, and reactive surface chemistry which allow very high activation degrees under harsh conditions even when starting from low-quality carbon precursors (i.e., high ash content, low density, low carbon content). Moreover, the features of the synthesized materials and their performance for the adsorption of target pharmaceutical compounds allow to envisage application in water treatment technologies. The apparent density of the materials will favor their separation by settling and the high volume of supermicropores associated with some mesopores will enable a good performance in competitive scenarios such as those found in wastewater treatment plants [8,10,51].

## 4. Conclusions

The set of data gathered in this study demonstrates that steam activation of acid-chars is an alternative methodology to KOH activation at high weight ratios since it also enables to synthesize superactivated carbons. In fact, the methodology followed in this study allowed to obtain activated carbons with textural properties resembling those of KOH activated acid-chars [18,21] or hydrochars [22,23,52], through the steam activation of sisal- and glucose-derived acid-chars. Regardless of the precursor (sisal or glucose), the use of 13.5 M H_2_SO_4_ concentration was determinant to prepare carbons with specific surface areas, reaching almost 2000 m^2^ g^−1^, presenting a nearly exclusive supermicropore network, and attaining total pore volumes close to 0.9 cm^3^ g^−1^.

Besides the change in the textural properties (from nonporous acid-chars to highly porous carbons) the characterization data also demonstrated that the steam activation resulted in changes in the surface chemistry (from acid solids to almost neutral carbons) and in the degree of aromatization that, as expected, increased upon activation.

The high performance for ibuprofen and iopamidol removal in both kinetic and equilibrium assays allows to consider their application for water purification purposes.

The overall results, prove the value of the proposed methodology (acid-mediated carbonization followed by steam activation) for processing a large range of biomass residues including those less prone to conventional carbonization routes (i.e., liquid precursors, as biomass with high moisture content and/or high inorganic content).

## Figures and Tables

**Figure 1 nanomaterials-12-03480-f001:**
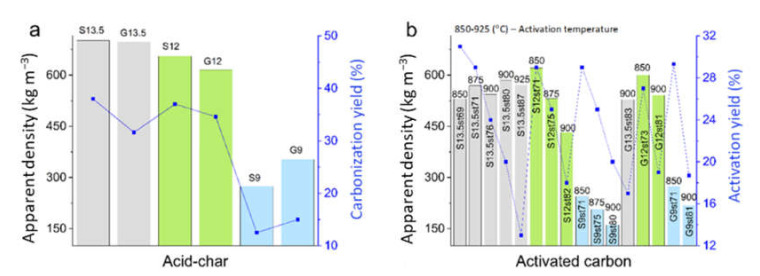
Influence of the synthesis conditions of acid-chars (**a**) and activated carbons (**b**) on, respectively, carbonization and activation yield, and apparent density.

**Figure 2 nanomaterials-12-03480-f002:**
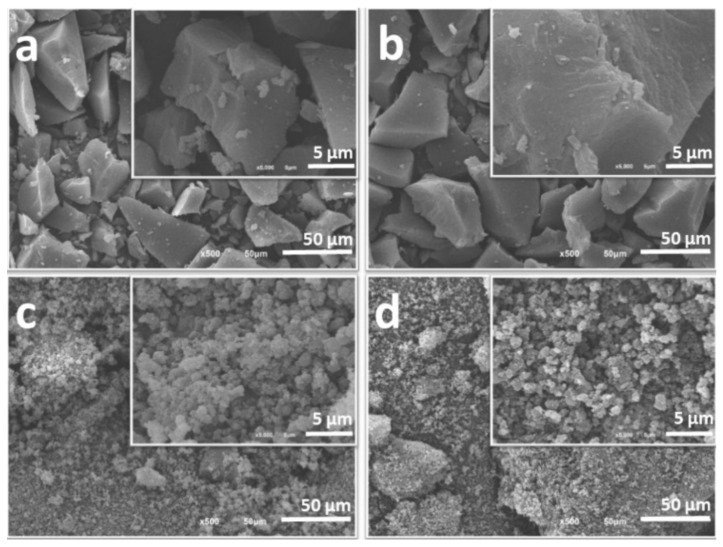
SEM images of activated carbons: (**a**) S13.5st87, (**b**) S12st82, (**c**) S9st80, (**d**) G9st81. Insets show the high magnification image of the carbon surface.

**Figure 3 nanomaterials-12-03480-f003:**
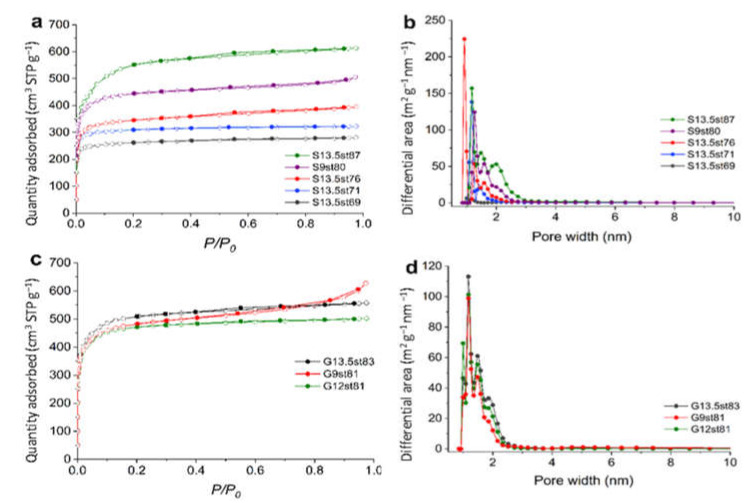
N_2_ adsorption–desorption isotherms and the pore size distribution of sisal- (**a**,**b**) and glucose-derived (**c**,**d**) activated carbon samples.

**Figure 4 nanomaterials-12-03480-f004:**
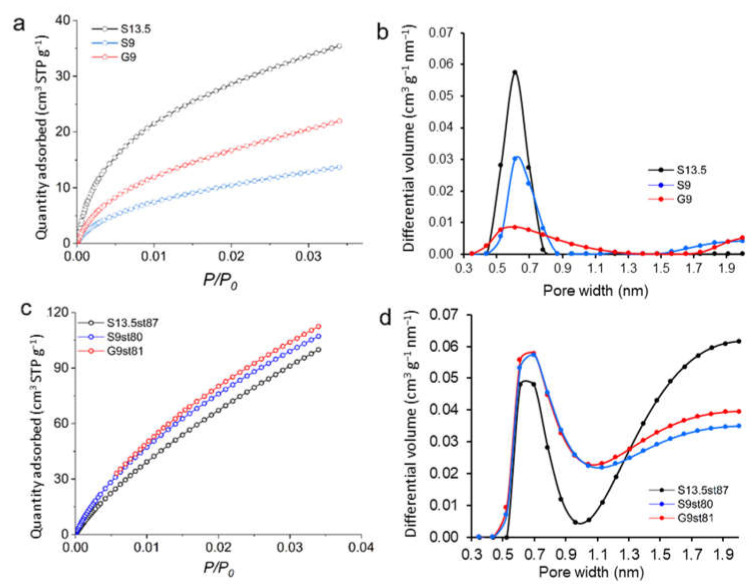
CO_2_ adsorption isotherms (**a**,**c**) and pore size distributions (**b**,**d**) of acid-chars (**a**,**b**) and activated carbons (**c**,**d**) at 0 °C.

**Figure 5 nanomaterials-12-03480-f005:**
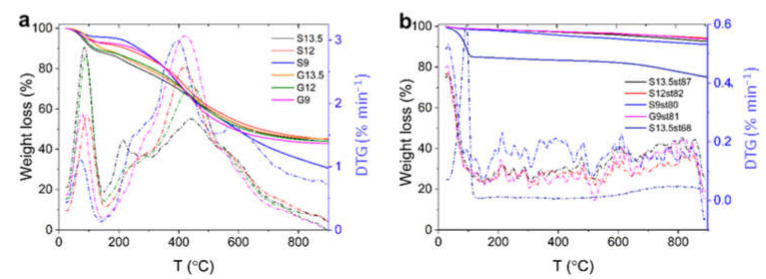
TG and DTG curves of acid-chars (**a**) and derived activated carbons (**b**) in N_2_ atmosphere.

**Figure 6 nanomaterials-12-03480-f006:**
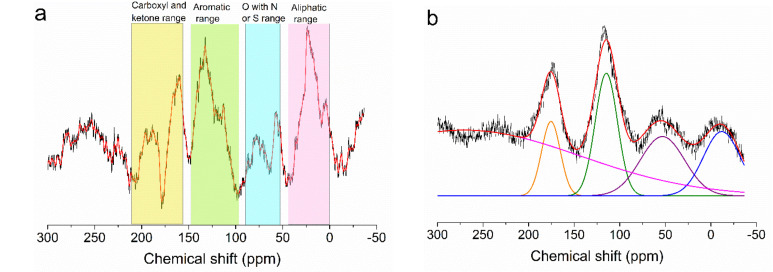
^13^C MAS NMR spectra of the initial acid-char (S13.5) (**a**) and derived activated carbon (S13.5st68) (**b**) samples.

**Figure 7 nanomaterials-12-03480-f007:**
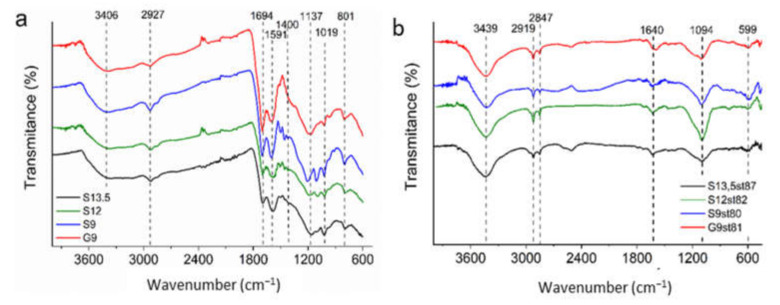
FTIR spectra of acid-chars (**a**) and derived activated carbons (**b**) samples.

**Figure 8 nanomaterials-12-03480-f008:**
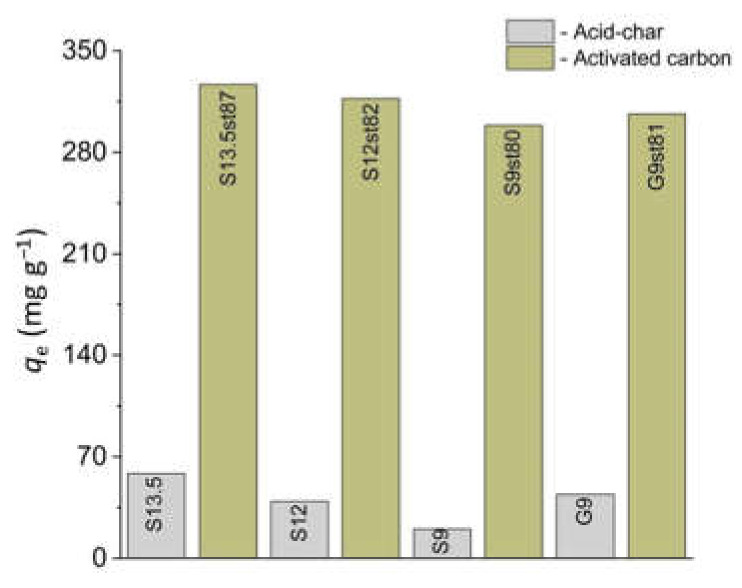
Ibuprofen uptake on acid-chars and activated carbon samples at pH 5.5 (experimental conditions: temperature 30 °C, initial concentration 120 mg L^−1^, volume 9 mL, weight of adsorbent: 12 mg for acid-chars and 6 mg for activated carbons).

**Figure 9 nanomaterials-12-03480-f009:**
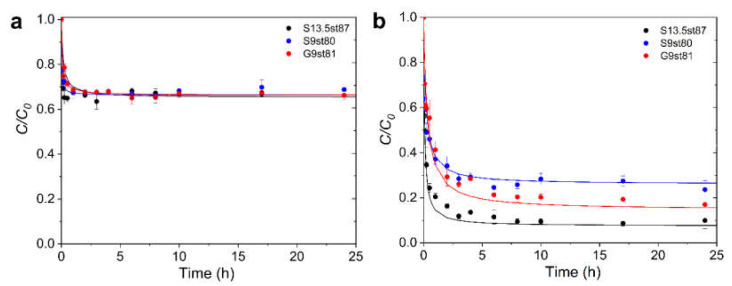
Adsorption kinetics of ibuprofen (**a**) and iopamidol (**b**) onto activated carbon samples (experimental conditions: temperature 30 °C, initial concentration 180 mg L^−1^, volume 30 mL, and weight of activated carbon 6 mg).

**Figure 10 nanomaterials-12-03480-f010:**
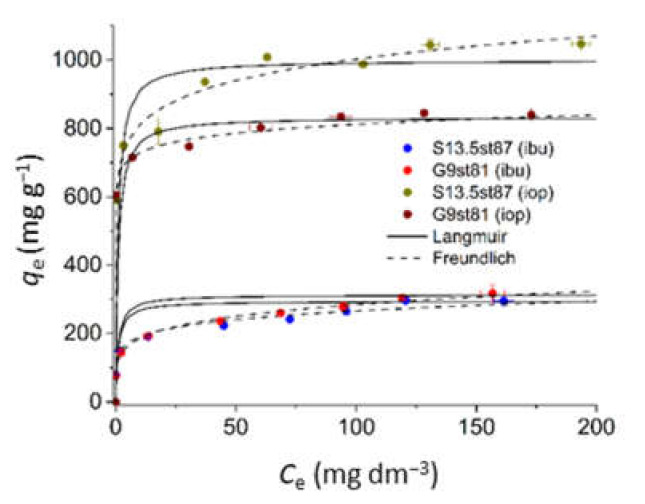
Adsorption isotherms of ibuprofen and iopamidol onto activated carbons (experimental conditions: pH 5.5; initial concentration 30–400 mg∙L^−1^, and 0.2 g L^−1^ of activated carbon).

**Table 1 nanomaterials-12-03480-t001:** Elemental analysis and pH_PZC_ values of acid-chars and derived activated carbons.

Sample	C (wt.%)	H (wt.%)	N (wt.%)	S (wt.%)	O (wt.%)	pH_PZC_
S13.5	60.43	3.91	0.39	1.17	34.10	2.2
S12	55.21	3.31	0.47	3.05	37.96	2.3
S9	60.38	4.14	0.39	1.16	33.98	2.3
G13.5	62.00	3.53	0.46	0.95	33.06	-
G12	57.13	3.53	0.26	2.59	36.49	-
G9	57.98	4.00	0.26	1.60	36.16	2.5
S13.5st87	-	-	-	-	-	6.1
S12st82	89.89	0.58	1.05	0.27	8.21	6.2
S9st80	93.91	0.10	0.96	0.31	4.72	5.7
G9st81	92.39	0.85	2.18	1.43	3.15	6.0

**Table 2 nanomaterials-12-03480-t002:** Textural properties of the activated carbon materials.

Sample	*A*_BET_ (m^2^ g^−1^)	*V*_total_ (cm^3^ g^−1^)	*V*_meso_ (cm^3^ g^−1^)	α_s_ Method		
				*V*_α total_ (cm^3^ g^−1^)	*V*_α ultra_ (cm^3^ g^−1^)	*V*_α super_ (cm^3^ g^−1^)
*Sisal-derived activated carbons*						
S13.5st87	1987	0.96	0.09	0.87	0.00	0.87
S13.5st80	1756	0.78	0.05	0.73	0.03	0.70
S13.5st76	1520	0.65	0.03	0.62	0.16	0.46
S13.5st71	1232	0.50	0.01	0.49	0.26	0.23
S13.5st69	1149	0.47	0.01	0.46	0.25	0.21
S12st82	1827	0.83	0.08	0.75	0.01	0.74
S12st75	1430	0.60	0.03	0.57	0.20	0.37
S12st71	1109	0.46	0.02	0.44	0.23	0.21
S9st80	1670	0.82	0.16	0.66	0.05	0.61
S9st75	1382	0.64	0.10	0.54	0.16	0.38
S9st71	1275	0.54	0.04	0.50	0.27	0.23
*Glucose-derived activated carbons*						
G13.5st83	1910	0.87	0.08	0.79	0.00	0.79
G12st86	1899	0.89	0.09	0.80	0.00	0.80
G12st81	1787	0.78	0.04	0.74	0.01	0.73
G12st73	1158	0.47	0.01	0.46	0.23	0.23
G9st81	1697	0.91	0.28	0.63	0.05	0.58
G9st71	1240	0.58	0.11	0.47	0.24	0.23

*V*_α total_—total micropore volume, *V*_α ultra_—volume of ultramicropores (width less than 0.7 nm), *V*_α super_—volume of supermicropores (width between 0.7–2 nm), and *V*_meso_—difference between *V*_total_ and *V*_α total_.

**Table 3 nanomaterials-12-03480-t003:** Physico-chemical properties of ibuprofen and iopamidol (adapted from ref. [23] with permission of the Royal Society of Chemistry).

	Ibuprofen (IBU)	Iopamidol (IOP)
Chemical structure	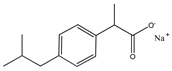	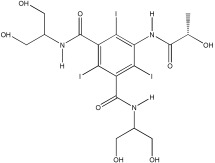
pKa	-	10.7
Water solubility (mg mL^−1^)	100	>200,000
log Ko/w	3.97	2.42
Molecular dimensions (nm)	1.32 (length) × 0.72 (width) × 0.72 (thickness)	Monomer 1.50 (length) × 1.50 (width) × 0.60 (thickness) Dimer 1.50 (length) × 1.50 (width) × 1.20 (thickness) Trimer 1.50 (length) × 1.50 (width) × 1.80 (thickness)

**Table 4 nanomaterials-12-03480-t004:** Pseudo-second order model parameters of the kinetic adsorption of ibuprofen and iopamidol onto the carbons obtained by steam activation of the sisal and glucose-derived acid-chars.

Sample	*k*_2_ (g mg^−1^ h^−1^)	*R* ^2^	*h* (mg g^−1^ h^−1^)	*t*_1/2_ (h)	Calculated Values			Experimental Results	
					*q*_e_,_calc_		*C*_e_,_calc_ (mg L^−1^)	*q*_e_,_exp_ (mg g^−1^)	*C*_e_,_exp_ (mg L^−1^)
					(mg g^−1^)	(mmol g^−1^) *			
*Ibuprofen*									
S13.5st87	0.021	0.9967	2000	0.16	313	1.53	118	327	117
S9st80	0.109	0.9975	10000	0.03	303	1.48	119	299	121
G9st81	0.036	0.9990	3333	0.09	303	1.48	119	306	118
*Iopamidol*									
S13.5st87	0.014	0.9999	10000	0.08	833	1.07	13	834	8
S9st80	0.008	0.9990	3333	0.20	667	0.86	47	706	38
G9st81	0.004	0.9996	2500	0.31	769	0.99	26	745	31

*k*_2_ is the pseudo-second order rate constant; h is the initial adsorption rate; *t*_1/2_ is the half-life time; *q*_e,calc_ is the adsorption capacity calculated by the kinetic model and * calculated considering the molecular weight of ibuprofen anion and iopamidol monomer; *C*_e_,_calc_, is the equilibrium concentration of analytes remaining in solution after adsorption; and *q*_e,exp_ is the adsorption capacity experimentally obtained.

**Table 5 nanomaterials-12-03480-t005:** Parameters of the equilibrium isotherm models of activated carbon samples.

Parameters	S13.5st87		G9st81	
	IOP	IBU	IOP	IBU
*Langmuir equation*				
qm (mg g^−1^)	1111	294	833	323
KL (dm^3^ mg^−1^)	0.29	0.16	3.00	0.13
R^2^	0.9987	0.9850	0.9988	0.9879
χ^2^	33.22	53.46	6.33	83.83
*Freundlich equation*				
1/nF	0.0928	0.2063	0.0587	0.0823
KF (mg^1−1/n^(L)^1/n^ g^−1^)	654	105	619	179
R^2^	0.9639	0.8740	0.9244	0.9027
χ^2^	9.01	4.32	3.99	1.16

*χ*^2^ = $\sum \frac{{\left(q_{e}-q_{m}\right)}^{2}}{q_{m}}$ where *q_e_* is the experimental equilibrium uptake and *q_m_* is the equilibrium uptake calculated from the model [50].

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
