# Peer review of "Steam Activation of Acid-Chars for Enhanced Textural Properties and Pharmaceuticals Removal"

_nanomaterials, 2022, doi:10.3390/nano12193480_

Round 1

Reviewer 1 Report (Previous Reviewer 2)

(1) For “The polycondensation of both precursors were performed at different concentrations of H2SO4 (13.5M, 12M, 9M) at 90 oC during 6 h and, in the case of sisal (S),...”, would  sisal (Agave sisalana) residues be dehydrated by sulfuric acid (13.5M)?

(2) For equation(1), a "x" should be added before "100%";

(3) What is "thermal analysis with MS detection"?

(4)  "Before adsorption experiments the samples (50-60 mg) were outgassed overnight"? It's really energy-intensive.

(5) For "Adsorption kinetic and isotherm experiments were conducted to examine the mechanisms of pharmaceutical drugs adsorption on the synthesized carbon materials", characterization of adsorbents befor-, post-adsorption and desorption mught be more convincing. 

(6) "...the steam activation of the acid-chars prepared in this study allowed to prepare activated carbons with apparent densities similar to those reported for the K2CO3 activation of acid-chars S12 or S13.5, and higher than those obtained by KOH activation", is the adsorption capacity of activated carbon determined by only its density?

(7) From the SEM images of activated carbons, they are really rough and irregular.

(8) Table 3 is suggested to remove from the manuscript.

(9) Most recently publicated researches of biochars in Molecules and Nanomaterial are suggested to be cited and discussed.

Author Response

On behalf of all the authors I want to thank the reviewer for the careful reading of the manuscript and all the comments which allowed to greatly improve the manuscript.

(1) For “The polycondensation of both precursors were performed at different concentrations of H2SO4 (13.5M, 12M, 9M) at 90 oC during 6 h and, in the case of sisal (S),...”, would  sisal (Agave sisalana) residues be dehydrated by sulfuric acid (13.5M)?

Answer: In fact, what we have in this process is a quite complex polycondensation mechanism where dehydration of the saccharic units resulting from the acid degradation of the precursor components (e.g. cellulose and hemicellulose) must be one of the steps.

(2) For equation (1), a "x" should be added before "100%";

Answer:  We do not understand the comment because there is a ”x” before the 100 % in equation 1. This must be a formatting issue depending on the computer.                     

(3) What is "thermal analysis with MS detection"?

Answer: “MS” means Mass Spectroscopy, as it is now clear in the revised version.

(4)  "Before adsorption experiments the samples (50-60 mg) were outgassed overnight"? It's really energy-intensive.

Answer: The pre-treatment conditions for the adsorption assays depend on the material we want to study. Inorganic materials, like zeolites, are usually heated at 300 ºC for 2h but carbon materials with quite labile surface functionalities cannot be submitted to such high temperatures . So, to achieve a correct surface “cleaning” we use lower temperature but longer treatments, in accordance with the recommended best practices of the carbon materials community.

(5) For "Adsorption kinetic and isotherm experiments were conducted to examine the mechanisms of pharmaceutical drugs adsorption on the synthesized carbon materials", characterization of adsorbents befor-, post-adsorption and desorption mught be more convincing. 

Answer: We agree with the reviewer and, in fact, that is an aspect that we are exploring in other studies. In the case of the present work our objective was to complete a broader study focused on the preparation of activated carbons from acid-chars. As it is mentioned along the text, we previously used chemical activation with KOH and K2CO3 and in the present work steam activation was successfully explored.

(6) "...the steam activation of the acid-chars prepared in this study allowed to prepare activated carbons with apparent densities similar to those reported for the K2CO3 activation of acid-chars S12 or S13.5, and higher than those obtained by KOH activation", is the adsorption capacity of activated carbon determined by only its density?

Answer: To have a high adsorption capacity we must deeply destroy the compact carbonaceous structure of the char obtained in the first step of the preparation process (carbonization). The way by which that “destruction” is made determines the final characteristics of the material. What we “see” with our results is that with the steam activation the reaction of the activating agent with the carbonaceous matrix (to create the desired porosity) occurs in a more controlled way, that is, consuming a lesser amount of the carbonaceous skeleton resulting in more dense materials than those prepared by KOH activation. As we mention in the manuscript, the characteristics of the materials obtained are not commonly reported what is mist probably due to the highly reactive surface chemistry of the acid-chars used as precursors. Regarding the correlation between adsorption capacity and density it must be stressed that density is a relevant parameter when envisaging to assess volumetric adsorption data. High densities allow high packing and higher adsorption capacity per volume, lower occupancy of warehouse and lower volumes of saturated materials.

(7) From the SEM images of activated carbons, they are really rough and irregular.

Answer: We agree with the comment and the analysis of the SEM images was modified in accordance.

(8) Table 3 is suggested to remove from the manuscript.

Answer: Table 3 gathers some physico-chemical characteristics of the two pharmaceuticals used as probe molecules. We believe that this information is needed to understand the discussion of the liquid phase assays, so we prefer to keep the Table in the manuscript aiming for higher readability.

(9) Most recently publicated researches of biochars in Molecules and Nanomaterial are suggested to be cited and discussed.

Answer:  The initial part of the introduction was revised, and a new paragraph was added to introduce the problematic of wastewater treatment issues. Regarding the possible solutions adsorption-based processes and, obviously, activated carbons (importance and preparation strategies) were mentioned. Several references were introduced to support the sentences.

Reviewer 2 Report (New Reviewer)

This paper presents interesting data with possible practical application.

Some minor corrections are necessary, as follows:

1.       How it was chosen the dose of adsorbent?

2.       The parameters obtained from the pseudo-first kinetic order model should also be included in the table 4.

3. Please improve the discussion part comparing the adsorption capacity obtained with your material with literature data.

Author Response

On behalf of all the authors I want to thank the reviewer for the careful reading of the manuscript and all the comments which allowed to greatly improve the manuscript.

This paper presents interesting data with possible practical application.

Some minor corrections are necessary, as follows:

  1. How it was chosen the dose of adsorbent?

Answer: The dose of adsorbent was the same used in previous studies so we could compare the results obtained with the materials prepared with those already achieved with other carbon materials.

  1. The parameters obtained from the pseudo-first kinetic order model should also be included in the table 4.

Answer: As it is mentioned in the manuscript the fitting of the kinetic data to the pseudo-first order model was tested but due to the statistical parameter (R2) values obtained – see Table below – we consider that the kinetic parameters that could be obtained have no physical meaning. In the revised manuscript we present the highest value of R2 obtained with the data of each compound.

Pseudo-first order (R2)

Ibuprofen

S13.5st87

0.1295

S9st80

0.0992

G9st81

0.4308

Iopamidol

S13.5st87

0.5278

S9st80

0.5293

G9st81

0.7603

  1. Please improve the discussion part comparing the adsorption capacity obtained with your material with literature data.

Answer: Following the reviewer suggestion we added a paragraph comparing the behaviour of the samples regarding the adsorption of ibuprofen and iopamidol with the performance of two commercial samples we reported in previous publications.

Reviewer 3 Report (New Reviewer)

Biomass derived porous carbonaceous materials are promising materials for waste water treatment. The topic of manuscript is of broad interesting to readers. The experiments are well designed and the manuscript is well organized. Minor revision is suggested by solving the following issues.

1.     It would be better to add a paragraph to focus on the waste water treatment, especially on pharmaceuticals removal, with various absorbents and do some comparisons in the introduction part. Some typical references are suggested to be cited: Journal of Bioresources and Bioproducts 2020, 5 (4), 238-247; Chemical Engineering Journal, 2022, 427, 131749; Journal of Environmental Chemical Engineering, 2021, 9(1), 104885.

2.     Biomass is a promising precursor for porous carbon materials. Various carbonaceous products have been obtained from biomass via pyrolysis and activation. More references are suggested to be cited, for example Biochar 2022, 4 (1), 50; Journal of Bioresources and Bioproducts 2021, 6 (2), 142-151; Diamond and Related Materials 2022, 128, 109238.

3.     There are too many labels on X axials of Figure 4b and 4d. Please revise them.

Round 2

Reviewer 1 Report (Previous Reviewer 2)

(1) What is "thermal analysis with MS detection"? Answer: “MS” means Mass Spectroscopy, as it is now clear in the revised version. I do know MS means Mass Spectroscopy, how to perfom the characterization using thermal analysis along with MS detection? Is it TG-MS?

(2) For "Adsorption kinetic and isotherm experiments were conducted to examine the mechanisms of pharmaceutical drugs adsorption on the synthesized carbon materials", characterization of adsorbents befor-, post-adsorption and desorption mught be more convincing. At least Supporting Information should be provided rather than evading this comment.

(3) For Table 3, it presents normal information of common  reagents, it should be removed!

(4)  For “The polycondensation of both precursors were performed at different concentrations of H2SO4 (13.5M, 12M, 9M) at 90 oC during 6 h and, in the case of sisal (S),...”, would the sisal (Agave sisalana) residues be dehydrated by sulfuric acid (13.5M)? If dehydration occurred, the authors should provide some deep analyses rather than simply respond subjectively to the reviewer.

Author Response

See the file uploaded

Round 3

Reviewer 1 Report (Previous Reviewer 2)

It can be accepted now.

This manuscript is a resubmission of an earlier submission. The following is a list of the peer review reports and author responses from that submission.

Round 1

Reviewer 2 Report

(1) "ac-id-chars" or "acid-chars"? Be careful of the expressions.

(2)The acid catalyst (H2SO4 and/or H3PO4) was used to obtain oxygen-rich chars contain sulfur or phosphorus groups, what are the status of S and P existence? What are the methods to determine the change of state?

(3) Be careful of the the superscripts and subscripts;

(4) " X corresponds to the H2SO4 concentration (M) used ..."  was repeatedly used.

(5) High-concentration acid and very high temperature were applied in the preparation procedure, it seems to be environmentally unfriendly and high-energy consumption;

(6) "was mix with" should  be "was mixed with";

(7) What's the type of membrane filter of 0.45 µm pore size?

(8) Do acid-chars possess adsorption selectivity?

(9) From Fig. 2, the prepared samples are not nanomaterials!

(10) From  Fig.5, why are platforms of weight lose below 200 degree centigrade could be observed? They were prepared under very high temperatures.

(11) How were 13C MAS NMR spectra obtained? I don;t think the samples could be soluble in any solvent.

It should be rejected by the journal.